# Influence of Anesthesiology Protocol on the Quality of Intraoperative Nerve Monitoring During Thyroid Surgery, One-Year Single Center Experience

**DOI:** 10.3390/diagnostics14212351

**Published:** 2024-10-22

**Authors:** Marina Stojanovic, Milan Jovanovic, Matija Buzejic, Tanja Maravic, Branislav Rovcanin, Nikola Slijepcevic, Katarina Tausanovic, Vladan Zivaljevic

**Affiliations:** 1Medical Faculty, University of Belgrade, 11000 Belgrade, Serbia; marinailicstojanovic@gmail.com (M.S.); milanjovanovicceh@gmail.com (M.J.); maravic.tanja.tm@gmail.com (T.M.); dr.nikola.slijepcevic@gmail.com (N.S.); katarinatausanovic@gmail.com (K.T.); vladanzivaljevic@gmail.com (V.Z.); 2Center for Anesthesiology and Resuscitation, University Clinical Center of Serbia, 11000 Belgrade, Serbia; 3Clinic for Endocrine Surgery, University Clinical Centre of Serbia, Pasterova 2, 11000 Belgrade, Serbia

**Keywords:** muscle relaxant, intraoperative nerve monitoring, novel procedures, thyroid disease management

## Abstract

Background/Objectives: Anesthesia plays a very important role in the successful management of intraoperative neuromonitoring (IONM). The aim of our study was to investigate the impact of anesthesia induction and maintenance on the quality of signals during surgeries on the thyroid and parathyroid glands using neuromonitoring. Methods: The study included 72 patients who underwent surgery with IONM for one year. All the patients were intubated using a Glidescope videolaryngoscope with a hyperangulated blade. Two different approaches were used to facilitate intubation: succinylcholine-1 mg/kg and rocuronium bromide-0.3 mg/kg. For anesthesia maintenance, total intravenous anesthesia (TIVA) or combined anesthesia was used. Patients’ body movements during operations, as well as electromyography signals from the vagus and recurrent laryngeal nerves before resection, were recorded as V1 and R1. Results: Intraoperative unwanted movements were recorded in 25% of patients. Undesired movements were more frequently recorded in the TIVA group compared to the combined anesthesia group (*p* < 0.001) as well as in patients who received succinylcholine compared to patients who received rocuronium bromide (*p* = 0.028). Type of anesthesia maintenance as well as type of muscle relaxant did not affect the quality of recorded nerve signals. (*p* = 0.169 and *p* = 0.894, respectively). Conclusions: The type of muscle relaxant used significantly affects the occurrence of undesirable movements during thyroid surgery with IONM, while the type of anesthesia maintenance did not influence either the quality of the obtained signal or the occurrence of undesirable movements.

## 1. Introduction

Injury to the recurrent laryngeal nerve (RLN) represents a serious complication during thyroid and parathyroid gland surgeries, affecting patients’ quality of life. The incidence of temporary RLN paralysis after thyroid gland surgery ranges from 0.2 to 3.5%, while the incidence of permanent paralysis is significantly lower, ranging from 0 to 1.6%, respectively [1,2,3]. Intraoperative neural monitoring (IONM) is a widely accepted non-invasive technique aimed at identifying, better visualizing, and assessing nerve function during surgeries on the thyroid and parathyroid glands [4,5,6,7,8]. In recent years, it has become the standard of modern endocrine surgery.

An anesthesiologist plays a crucial role in surgeries involving neuromonitoring. Anesthesia specifics primarily relate to the type of tube that is used, correct depth and position of the tube, choice of muscle relaxant for induction, and maintenance of anesthesia [9,10,11]. Tube malposition during intubation or head hyperextension can lead to signal loss and incorrect information. The use of videolaryngoscopes has been shown to be superior to standard Macintosh laryngoscopes [12]. One of the most important decisions is which type of muscle relaxant to use and in what dose. Numerous protocols have been developed in recent years to provide optimal conditions for performing these surgeries [3].

The aim of our study was to determine whether the type of muscle relaxant and the method of anesthesia maintenance (total intravenous anesthesia (TIVA) or combined anesthesia) affect the quality of signals during surgeries on the thyroid and parathyroid glands using neuromonitoring.

## 2. Materials and Methods

The study was designed as an observational analysis of patients who underwent surgery between May 2023 and May 2024 at the Clinic for Endocrine Surgery, University Clinical Center of Serbia (UKCS). The written informed consent was obtained from each patient and the study protocol was approved by the Ethics Committee of Clinic for Endocrine Surgery UKCS (No 187/10).

The study included patients of both genders, with American Society of Anesthesiologists physical classification (ASA) I-III status, who underwent surgery using neuromonitoring (thyroidectomy, lobectomy, neck dissection, and reoperation). The indication for IONM was determined by the operating surgeon based on preoperative assessment. Due to the limited availability of resources and IONM systems, patients for IONM were selected based on demographic and clinical characteristics.

The study excluded non-elective patients, patients with a body mass index (BMI) over 35 kg/m^2^, as well as patients with incomplete documentation, significant comorbidities such as heart, kidney, or liver disease, psychiatric or neurological disorders, allergy to any anesthetic drug, and those who denied written informed consent.

### 2.1. Anesthesia Protocol

In the operating room, the electrocardiography (ECG), heart rate (HR), non-invasive blood pressure (NIBP), pulse oxygen saturation (SpO_2_), bispectral index (BIS, Covidien, Minneapolis, MN, USA) of the patients were monitored. For the induction of anesthesia, the following medications were used: fentanyl 1–2 mcg/kg I and propofol 2 mg/kg IV. Two different approaches were used to facilitate intubation: depolarizing and non-depolarizing muscle relaxants. Succinylcholine was used at a dose of 1 mg/kg of the actual body weight, while rocuronium bromide was used at a dose of 0.3 mg/kg of the ideal body weight. The decision regarding the type of muscle relaxant was made by the attending anesthesiologist.

All the patients were intubated using a Glidescope videolaryngoscope with a hyperangulated blade, with the head positioned in a neutral position. A special tube with surface electrodes embedded in an endotracheal tube was used, which was positioned between the vocal cords under direct visualization, as recommended by the manufacturer. The tube was secured in the midline of the face, on the forehead, to prevent errors in the subsequent monitoring due to tube rotation. After head extension, placing a pillow under the scapulae, and positioning the patient in a definitive position, the tube depth and potential rotation were checked and corrected by reusing the videolaryngoscope.

TIVA was used for anesthesia maintenance, utilizing TCI pumps (target-controlled infusion) for propofol and remifentanil at doses that provided an adequate depth of anesthesia. The depth of anesthesia was monitored using BIS and maintained within the range of 40–60. The patients were ventilated with a gas mixture of oxygen and air in a 50–50 ratio, at a flow rate of 2 L/min. Depending on the preferences of the attending anesthesiologist, in a certain number of patients, an inhalation anesthetic sevoflurane was also used in the gas mixture at a dose of 0.4–0.6 MAC (minimum alveolar concentration).

### 2.2. Neuromonitoring Protocol

For the identification of the vagal nerve and RLN, the Medtronic 3.0 IONM system with standard settings was used. During surgery, the vagal nerve and recurrent laryngeal nerve were identified both macroscopically and with IONM. The maximal amplitudes and latency were recorded.

Patient’s body movements during operations and electromyography signals from the vagus and recurrent laryngeal nerves before resection were recorded as V1 and R1. Intraoperative body movement was defined as the patient’s limb movement, swallowing, or coughing during the operation. The study investigated the impact of the type of muscle relaxant and anesthesia on the occurrence of patient movements as well as on the quality of the obtained signal. In the case of unwanted movements, a bolus of propofol was administered at a dose of 0.5 mg/kg. The quality of IONM was measured using the obtained V1 amplitude. Satisfactory and poor signal quality values were defined as V1 amplitude >500 μV and <500 μV, respectively, or R1 amplitude >500 μV and <500 μV. The frequency of postoperative nausea and vomiting was also recorded in the postoperative period.

### 2.3. Statistical Analysis

The SPSS 26.0 statistical software (IBM Corporation, Armonk, NY, USA) was used for analysis. According to the normality of the variables, which were tested using the Kolmogorov–Smirnov test, for normally distributed measurement data, results were expressed as $\overset{-}{x}$ ± SD, and for non-normally distributed data, the median [quartile range] was used. Categorical nominal variables were analyzed using the chi-square or Fisher exact test. Statistical significance was set at *p* < 0.05 for all comparisons.

## 3. Results

The study included 72 patients. Table 1 shows the basic preoperative characteristics of the patients under investigation. The majority of participants were female (77.8%), with an average age of 51 years. More than half of the patients had some associated comorbidities (ASA II 59.7%, ASA III 30.6%), while 8.3% of patients underwent reoperation. Based on preoperative otolaryngoscopic findings, one patient had paralysis and two had recurrent nerve paresis. The average size of the dominant nodule was 28 ± 14.8 mm, with a minimum recorded value of 5 mm and a maximum of 70 mm.

Table 2 shows the intraoperative characteristics. The most commonly performed operation was total thyroidectomy (65.28%), as presented in Figure 1. Neuromonitoring was used during parathyroidectomy in three patients and during thyroidectomy with parathyroidectomy in 4.17% of cases. The average duration of the operation was 67.5 min, while the average duration of anesthesia was 82.5 min. More than half of the patients received methylprednisolone intraoperatively (54.2%), while calcium gluconate was administered intraoperatively in 23.6% of patients. Total intravenous anesthesia was used for maintaining anesthesia in 33 patients (45.8%), while a combination of intravenous and inhalation anesthetics was used in 54.2% of cases. The majority of patients received a non-depolarizing muscle relaxant at the induction of anesthesia (61.1%), while succinylcholine was used in 38.9% of patients. Intraoperative unwanted movements were recorded in 25% of patients, and postoperative nausea and vomiting occurred in 8.3% of the cases. Table 2 also displays the average amplitudes on the vagus and recurrent nerves before dissection.

An amplitude on the vagus nerve of less than 500 μV before dissection was recorded in 28 patients, as described in Table 3. Among patients who received TIVA, 10 patients had amplitudes less than 500 μV, compared to 18 patients in the group receiving combined anesthesia. There was no statistically significant difference observed (*p* > 0.05). The amplitude on the recurrent laryngeal nerve less than 500 μV was observed in eighteen patients, with six in the TIVA group and twelve in the combined anesthesia group, with no statistically significant difference found (*p* > 0.05). When comparing the influence of different muscle relaxants on the amplitude value, no statistically significant difference was observed for either vagus nerve amplitude (*p* = 0.894) or recurrent laryngeal nerve amplitude (*p* = 0.118). Undesired movements were more frequently recorded in the TIVA group: fourteen out of nineteen patients, compared to four out of thirty-nine patients in the combined anesthesia group. A high statistical significance was observed (*p* = 0.000). Additionally, undesired movements were more frequently recorded in the group of patients who received succinylcholine compared to patients who received rocuronium bromide (10 out of 18 vs. 8 out of 36), with a statistically significant difference noted (*p* = 0.028).

## 4. Discussion

The results of our study showed that the intensity of amplitude on the vagus and recurrent laryngeal nerves before dissection was equally good and there was no difference based on the type of muscle relaxant used. Additionally, the use of inhalation anesthetics (sevoflurane) did not affect the signal quality compared to TIVA. However, we demonstrated that unwanted movements were more frequently recorded in the group of patients who received depolarizing muscle relaxants during anesthesia induction, as well as in patients whose anesthesia was maintained using intravenous anesthetics.

Numerous studies have demonstrated the superiority of videolaryngoscopy over direct laryngoscopy, especially in the context of using IONM in thyroid surgery [13,14,15]. The study by Kriege et al. also showed that, in addition to better visibility of the glottis and improved tube positioning, the success rate for first-attempt intubation was significantly higher with videolaryngoscopy compared to direct laryngoscopy (96% vs. 66%, *p* < 0.001). Simultaneously, it was observed that the time required for adequate visualization of the glottis was shorter in the videolaryngoscopy group, although the time needed for tube placement was not significantly different (*p* = 0.008, *p* = 0.13) [16]. Additionally, it was found that inadequate electromyographic signals (<500 mV) were significantly higher (27%) in the direct laryngoscopy group compared to the videolaryngoscopy group (9%) [16]. Most anesthesiologists opt for using a videolaryngoscope as a secondary option after the first unsuccessful attempt with a classic Macintosh laryngoscope. When it comes to thyroid surgery, especially when IONM is used, the use of videolaryngoscopy has numerous advantages, as previously mentioned studies have shown [17,18]. For this reason, all the patients in our study were intubated using a videolaryngoscope with a hyperangulated blade. In contrast to our study where the patients were intubated with their head in a neutral position and later rechecked for tube position after definitive positioning, in the study by Won et al., patients were positioned in the thyroid surgical posture and better glottis visualization was demonstrated with the use of videolaryngoscopy [12]. The most delicate decision concerns the type and dose of muscle relaxant that will be prescribed for the purpose of facilitating intubation. Numerous studies have proposed different protocols, such as intubation without muscle relaxants, using a depolarizing relaxant, as well as using various doses of non-depolarizing relaxants with and without sugammadex. The study by Yang et al. compared two doses of rocuronium bromide (0.3 mg/kg and 0.6 mg/kg). It was shown that the group of patients who received a higher intubation dose of rocuronium bromide had a significantly higher incidence of poor (<100 µV) V1 signals (20.7% vs. 6.2%). After the administration of 2 mg/kg of sugammadex, the quality of the signal was over 99% in both groups, but the incidence of unwanted muscle movements during the operation was much lower in the group with the higher intubation dose of rocuronium bromide (0 vs. 16) [19].

Gunes et al. in their research using TOF (train-of-four) monitoring demonstrated that the time required for the complete recovery of laryngeal electromyography after the administration of sugammadex (2 mg/kg) was 26.07 ± 3.26 min compared to 50.0 ± 8.46 min in patients who did not receive sugammadex. The intubation dose of rocuronium was the same and amounted to 0.6 mg/kg (*p* < 0.001). The train-of-four ratio recovered from 0 to >0.9 within 4 min [20].

Different doses, primarily of rocuronium bromide, have been proposed as possible options for the use of IONM. Lower doses of 0.3 mg/kg, followed by 0.6 mg/kg and up to 0.9 mg/kg, have been considered. A lower dose of 0.3 mg/kg can provide the adequate conditions for intubation and satisfy the needs of neuromonitoring; however, it has been shown that the incidence of unwanted movements during surgery is significantly higher compared to when larger doses are used [21]. This is particularly relevant in robotic surgery, where the occurrence of unwanted movements can significantly impact tissue injury and other undesirable complications.

Lan et al. in their prospective, randomized study used different intubation doses of rocuronium bromide and showed that the incidence of intraoperative body movement between the groups receiving 0.6 to 0.9 mg/kg was statistically significant compared to the group receiving 0.3 mg/kg (both *p* < 0.001) [22].

Marusch et al. demonstrated through the use of accelerometry that there is a significant difference in the degree of relaxation between the adductor pollicis muscle and the vocalis muscle. The laryngeal muscles exhibited a shorter response time compared to the adductor pollicis and recovered more quickly, which justifies the use of neuromonitoring during thyroid surgery [23]. It has also been shown that a low intubation dose of rocuronium, compared to no muscle relaxant, does not compromise the IONM signal prior to dissection and improves the intubation condition compared to a strategy without relaxants [21]. This was one of the reasons why we used lower recommended doses of non-depolarizing muscle relaxants in our study. In recent years, the use of sugammadex after rocuronium has become very popular as an effective and precise neuromonitoring regimen. Different protocols have been used regarding the dosage of sugammadex, ranging from 0.25 mg/kg to 2 mg/kg [24,25,26,27,28]. It has been shown that lower doses can improve the recovery of the neuromuscular junction, while higher doses increase the incidence of undesirable movements during surgeries. The importance of the precise timing of sugammadex administration has also been highlighted. Several protocols have been proposed, such as immediately after intubation and fixation of the tube, during skin incision, 10 min after skin excision, and during the identification of the vagus nerve. The average time from the administration of sugammadex to V1 stimulation in published studies ranged from 3 to 32 min [4,29,30]. The most important limiting factor regarding sugammadex is still its high cost and the fact that it is not registered for use in this specific type of anesthesia, as is the case in our country. For this reason, sugammadex could not be used in our study; instead, we used low doses of rocuronium, with certain clinical limitations, which we compared with succinylcholine.

The research has several limitations. First of all, it is a retrospective analysis and lacks randomization. Due to limited financial resources, we were unable to include a larger number of patients. Additionally, due to the inability to monitor the degree of neuromuscular block, TOF was not used; we had to use lower doses of rocuronium bromide or the less popular succinylcholine, which is known to have numerous side effects and is no longer recommended as a drug of choice in modern anesthesiology practice, even in cases of thyroid surgery and high incidence of difficult intubation.

In conclusion, the results of our study confirm that the method of conducting anesthesia is a very important factor for the success of neuromonitoring in thyroid surgery. The type of muscle relaxant used significantly affects the occurrence of undesirable movements, while the type of anesthesia maintenance does not influence either the quality of the obtained signal or the occurrence of undesirable movements.

## 5. Conclusions

The type of muscle relaxant used significantly affects the occurrence of undesirable movements during thyroid surgery with IONM, while the type of anesthesia maintenance does not influence either the quality of the obtained signal or the occurrence of undesirable movements.

## Figures and Tables

**Figure 1 diagnostics-14-02351-f001:**
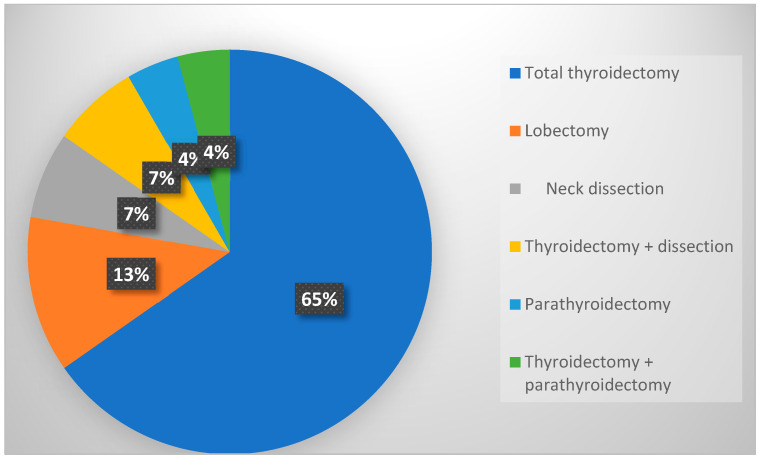
Type of surgery.

**Table 1 diagnostics-14-02351-t001:** Patient characteristics.

Total	N–72
Gender	
Female	56 (77.8%)
Male	16 (22.2%)
ASA status	
I	7 (9.7%)
II	43 (59.7%)
III	22 (30.6%)
Age (years)	51.5 ± 15.3
Reoperation status	6 (8.3%)
Preoperative ENT findings	
Paresis	1 (1.4%)
Paralysis	2 (2.8%)
Normal findings	69 (95.8%)
Bethesda classification	
1	4 (11.4%)
2	11 (31.4%)
3	6 (17.1%)
4	7 (20.0%)
5	6 (17.1%)
6	1 (2.9%)
Size of the nodules (mm)	28 ± 14.8 (Min–5 Max–70)

N—total number of patients; ASA—American Society of Anesthesiologists; ENT—ear, nose, and throat examination.

**Table 2 diagnostics-14-02351-t002:** Intraoperative characteristics.

The Variables	N (%)
Duration of surgery in minutes	82.1 ± 45.6: med–67.5
Duration of anesthesia in minutes	93.1 ± 45.7: med–82.5
Metylprednisolone intraoperatively	39 (54.2%)
Calcium intraoperatively	17 (23.6%)
Type of anesthesia:	
TIVA	33 (45.8%)
Combined anesthesia	39 (54.2%)
Type of muscle relaxant:	
Succinylcholine	28 (38.9%)
Rocuronium bromide	44 (61.1%)
Unwanted movements	18 (25.0%)
PONV	6 (8.3%)
Vagus nerve amplitude before dissection	680 ± 431.7 (162–2534)
Recurrent nerve amplitude before dissection	911.1 ± 578.9 (312–3248)

N—number of patients; TIVA—total intravenous anesthesia; PONV—postoperative nausea and vomiting.

**Table 3 diagnostics-14-02351-t003:** Intraoperative neuromonitoring.

		V1 < 500 μV	V1 > 500 μV	*p* Value	R1 < 500 μV	R1 > 500 μV	*p* Value	Movements, Yes	*p* Value
Type of anesthesia	TIVA	10 (30, 3%)	23 (69, 7%)	0.169	6 (18, 18%)	27 (81, 82%)	0.219	14 (42, 42%)	0.001
	Combined	18 (46, 15%)	21 (53, 85%)		12 (30, 77%)	27 (60, 33%)		4 (10, 26%)	
Muscle relaxant	Succinylcholine	10 (35, 71%)	23 (64, 29%)	0.894	8 (28, 57%)	20 (71, 43%)	0.118	10 (35, 71%)	0.028
	Rocuronium	14 (31, 82%)	30 (68, 18%)		6 (13, 64%)	38 (86, 36%)		8 (18, 18%)	

TIVA—total intravenous anesthesia; V1—vagus nerve amplitude before dissection; R1—recurrent nerve amplitude before dissection.

## Data Availability

The original contributions presented in the study are included in the article, further inquiries can be directed to the corresponding author.

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
