# Peer review of "Influence of Anesthesiology Protocol on the Quality of Intraoperative Nerve Monitoring During Thyroid Surgery, One-Year Single Center Experience"

_diagnostics, 2024, doi:10.3390/diagnostics14212351_

Round 1

Reviewer 1 Report

Comments and Suggestions for Authors

This article is a study of the use of intraoperative neuromonitoring (IONM) in thyroid surgery, examining the impact of anaesthetic protocols on the quality of the neuromonitoring signal. The study was conducted over a one-year period and is a summary of single-centre experience. The aim of this study was to determine whether the type of muscle relaxant and the method of anesthesia maintenance (total intravenus anesthesia-TIVA or combined anesthesia) affect the quality of signals during surgeries on the thyroid and parathyroid glands using neuromonitoring.

The manuscript describe details of the methods, results and conclusions of the study, using tables to present the data clearly. Terminology is used appropriately and statistics are reported clearly. IONM assessing nerve function during surgeries on the thyroid and parathyroid glands is an important topic, due to the risk of injury to the recurrent laryngeal nerve in these procedures. The effect of anaesthetic protocols on the quality of the IONM signal is a relevant and important area of study. The manuscript presented in a well-structured manner. The cited references are mostly recent publications (within the last 5 years) and relevant. Overall, the experimental design of this article is consistent with the hypothesis proposed by the test.

Lower doses of rocuronium bromide or the less popular succinylcholine, which is known to have numerous side effects and is no longer recommended as a drug of choice in modern anesthesiology practice, even in cases of thyroid surgery and high incidence of difficult intubation. Study should use more first-line inotropic drugs. And the choice of muscarinic medication was chosen by the attending surgeon, in such a way that randomisation was not carried out, and whether the attending surgeon's intraoperative manoeuvres had an impact on the study is uncertain.

The specific steps of the study were described in sufficient detail and the way the data was recorded was sufficiently precise, but the fact that the study was a single-centre study and that randomisation was not carried out, both of which would have affected the study's results reproducible.

The tables are expected to be clear and easy to interpret, given that they summarize numerical data in a structured format. The presentation of the results can be considered with more visual images.

Author Response

This article is a study of the use of intraoperative neuromonitoring (IONM) in thyroid surgery, examining the impact of anaesthetic protocols on the quality of the neuromonitoring signal. The study was conducted over a one-year period and is a summary of single-centre experience. The aim of this study was to determine whether the type of muscle relaxant and the method of anesthesia maintenance (total intravenus anesthesia-TIVA or combined anesthesia) affect the quality of signals during surgeries on the thyroid and parathyroid glands using neuromonitoring.

The manuscript describe details of the methods, results and conclusions of the study, using tables to present the data clearly. Terminology is used appropriately and statistics are reported clearly. IONM assessing nerve function during surgeries on the thyroid and parathyroid glands is an important topic, due to the risk of injury to the recurrent laryngeal nerve in these procedures. The effect of anaesthetic protocols on the quality of the IONM signal is a relevant and important area of study. The manuscript presented in a well-structured manner. The cited references are mostly recent publications (within the last 5 years) and relevant. Overall, the experimental design of this article is consistent with the hypothesis proposed by the test.

Response: Thank you for your comment and for summarizing the results in such a nice way.

Lower doses of rocuronium bromide or the less popular succinylcholine, which is known to have numerous side effects and is no longer recommended as a drug of choice in modern anesthesiology practice, even in cases of thyroid surgery and high incidence of difficult intubation. Study should use more first-line inotropic drugs. And the choice of muscarinic medication was chosen by the attending surgeon, in such a way that randomisation was not carried out, and whether the attending surgeon's intraoperative manoeuvres had an impact on the study is uncertain.

Response: Thank you for the comment. Muscarinic medication was not used in our study and surgeons did not have an impact on anesthesia protocol in such a way.

The specific steps of the study were described in sufficient detail and the way the data was recorded was sufficiently precise, but the fact that the study was a single-centre study and that randomisation was not carried out, both of which would have affected the study's results reproducible.

Response: Yes, thank you for noticing, this was evaluated and stated as one of limitations.

The tables are expected to be clear and easy to interpret, given that they summarize numerical data in a structured format. The presentation of the results can be considered with more visual images.

Response: Thank you for your suggestion, we added Figure 1, and also the Table 3 is rewritten.

Reviewer 2 Report

Comments and Suggestions for Authors

"Injury to the recurrent laryngeal nerve (RLN) represents one of the most serious 34 complications during thyroid and parathyroid gland surgeries."

I disagree :only bilateral palsy is  most serious , Hipoparathyroisdms permanent is also  a bad   complication . If is your opinion , please disccus it in "the discussion"

"For identification of vagal nerve and RLN Medtronic 3.0 IONM system with stand-96 ard setting was used. At the beginning of surgery, the vagal nerve and recurrent laryngeal 97 nerve were identified both macroscopic and with IONM. The maximal amplitudes and 98 latency were recorded"

Have you dissected or isolated the vagus nerve ? It is not necessary in the appproaches . I guessa you have to  carify this point 

Author Response

"Injury to the recurrent laryngeal nerve (RLN) represents one of the most serious 34 complications during thyroid and parathyroid gland surgeries."

I disagree :only bilateral palsy is  most serious , Hipoparathyroisdms permanent is also  a bad   complication . If is your opinion , please disccus it in "the discussion"

Response: Dear reviewer, thank you for the comment. We agree bilateral palsy is one of the most severe specific complications of thyroid surgery. Also, some debate that permanent hypoparthyroidism is even worse. I think we can agree that unilateral paralysis of RLN  follows after that. Taking your comment into account we have omitted the word most form the sentence.

"For identification of vagal nerve and RLN Medtronic 3.0 IONM system with stand-96 ard setting was used. At the beginning of surgery, the vagal nerve and recurrent laryngeal 97 nerve were identified both macroscopic and with IONM. The maximal amplitudes and 98 latency were recorded"

Have you dissected or isolated the vagus nerve ? It is not necessary in the appproaches . I guessa you have to  carify this point 

Response: Thank you for your comment. Standard operative procedure by the book is to first identify vagal nerve before and after dissection (lobectomy) of both thyroid lobes. We agree that in some cases in practice this is not always the case, but for the purpose of this study we strictly implemented this protocol. We rephrased the sentence to reflect this.

Round 2

Reviewer 2 Report

Comments and Suggestions for Authors

the placements have been arranged; the tables and demonstrations seem to be clearer (in the presented version the corrections appear which leave me a little confused); thus I think the work can be published